# Synthetic Flavonoid BrCl-Flav—An Alternative Solution to Combat ESKAPE Pathogens

**DOI:** 10.3390/antibiotics11101389

**Published:** 2022-10-11

**Authors:** Cristina-Veronica Moldovan, Mihaela Savu, Elodie Dussert, Haïrati Aboubacar, Laura Gabriela Sarbu, Simona Matiut, Benoit Cudennec, François Krier, Rozenn Ravallec, Lucian Mihail Birsa, Marius Stefan

**Affiliations:** 1Department of Biology, Faculty of Biology, The Alexandru Ioan Cuza University of Iasi, Bd. Carol I, Nr. 11, 700506 Iasi, Romania; 2UMR-T 1158, BioEcoAgro, University of Lille, 59650 Lille, France; 3Faculty of Chemistry, The Alexandru Ioan Cuza University of Iasi, Bd. Carol I, Nr. 11, 700506 Iasi, Romania; 4Investigații Medicale Praxis, Stradela Moara de Vânt, Nr. 35, 700377 Iasi, Romania

**Keywords:** synthetic flavonoid, antibacterial, ESKAPE pathogens, anti-biofilm activity, synergistic effect, low cytotoxicity, pro-inflammatory effect

## Abstract

ESKAPE pathogens are considered as global threats to human health. The discovery of new molecules for which these pathogens have not yet developed resistance is a high medical priority. Synthetic flavonoids are good candidates for developing new antimicrobials. Therefore, we report here the potent in vitro antibacterial activity of BrCl-flav, a representative of a new class of synthetic tricyclic flavonoids. Minimum inhibitory/bactericidal concentration, time kill and biofilm formation assays were employed to evaluate the antibacterial potential of BrCl-flav. The mechanism of action was investigated using fluorescence and scanning electron microscopy. A checkerboard assay was used to study the effect of the tested compound in combination with antibiotics. Our results showed that BrCl-flav displayed important inhibitory activity against all tested clinical isolates, with MICs ranging between 0.24 and 125 µg/mL. A total kill effect was recorded after only 1 h of exposing *Enterococcus faecium* cells to BrCl-flav. Additionally, BrCl-flav displayed important biofilm disruption potential against *Acinetobacter baumannii*. Those effects were induced by membrane integrity damage. BrCl-flav expressed synergistic activity in combination with penicillin against a MRSA strain. Based on the potent antibacterial activity, low cytotoxicity and pro-inflammatory effect, BrCl-flav has good potential for developing new effective drugs against ESKAPE pathogens.

## 1. Introduction

Antimicrobial resistance (AMR) remains a major public concern, posing a serious threat to human health and economic development around the world. According to the World Health Organization, antimicrobial resistance is one of the 10 global public health menaces facing humanity today, which increases mortality and morbidity and strains healthcare systems [1]. AMR is defined by the European Centre for Disease Prevention and Control (ECDC) as the ability of microorganisms (viruses, bacteria, fungi and parasites) to resist the action of one or more antimicrobial agents [2]. It may occur when antimicrobial drugs used to treat infections become less effective or inefficient due to changes in pathogenic microorganisms, especially bacteria. Extensive previous studies have estimated the AMR consequences in terms of deaths, hospital length of stay and healthcare costs [1,2,3,4,5]. New data from a 2022 study show the true AMR burden, with an estimated 4.95 million deaths associated with bacterial AMR in 2019, including 1.27 million deaths as a direct result of antibiotic-resistant bacterial infections [6]. One pathogen alone—methicillin-resistant *Staphylococcus aureus* (MRSA), a member of the ESKAPE group—was responsible for more than 100,000 deaths attributable to AMR in 2019 [6].

The ESKAPE group includes six highly virulent and antibiotic-resistant life-threatening pathogens: *Enterococcus faecium*, *Staphylococcus aureus*, *Klebsiella pneumoniae*, *Acinetobacter baumannii*, *Pseudomonas aeruginosa* and *Enterobacter* spp. [7]. On the other hand, SPEAKS pathogens (*S. aureus*, *P. aeruginosa*, *Escherichia coli*, *A. baumannii*, *K. pneumoniae* and *S. pneumoniae*) are the leading causes of AMR deaths [8]. Most of them exhibit multidrug resistance (MDR), which is considered one of the greatest challenges in clinical practice [9]. MDR is caused by the inappropriate use of antimicrobials, excessive drug usage, inadequate sanitary conditions and poor infection prevention and control practices [10]. The main mechanisms of ESKAPE MDR include drug inactivation/alteration, modification of drug-binding targets, reduced intracellular drug accumulation due to decreased outer membrane permeability or efflux pump activity and biofilm formation [11,12]. ESKAPE pathogens are known as biofilm producers [13,14]. This capability makes them resistant to antibiotics, as interior cells in a biofilm are protected from the effects of antimicrobial drugs or from host immune system actions [15]. Consequently, biofilm-forming bacteria could develop up to 1000-fold or more antibiotic resistance compared to planktonic cells [16].

More and more patients with ESKAPE infections are no longer responding to available treatments. Furthermore, when new antimicrobials come to market, bacteria are becoming quickly resistant to them. Several strategies were proposed to overcome MDR. Innovation of new effective drugs could offer solutions to minimize the impact of resistance to antibiotics. In addition, synergy and drug combinations are a successful strategy in fighting MDR bacteria and might help in prolonging the lifetime of current antibiotics [17]. Nevertheless, finding new innovative and effective solutions to keep pace with the antibiotic resistance developed by ESKAPE pathogens is particularly challenging and represents a crucial objective of current biomedical research [18].

In this context, flavonoids, a class of plant polyphenols, are potentially good candidates for developing new antimicrobial agents [19]. These natural compounds were used for centuries in traditional medicine to treat infectious diseases due to their antimicrobial, anti-inflammatory and antioxidant properties and represent a subject of intensive research. Synthetic flavonoids, however, could represent a more reliable source of new effective antimicrobial compounds due to their improved biological activities [20].

Our research has focused for several years on a new class of synthetic sulfur-containing tricyclic flavonoids with different halogen substituents at the benzopyran core [21,22], such as BrCl-flav (Figure 1). Although we previously showed that BrCl-flav is an effective antimicrobial compound with strong bactericidal and fungicidal effects against *S. aureus*, *E. coli* and *Candida* spp. [23,24], no investigations were carried out against pathogenic bacterial strains. Therefore, in the present study we focus not only on the antibacterial activity of BrCl-flav against antibiotic-resistant clinical isolates including ESKAPE pathogens, but also on biofilm formation and the effects of combinations with traditional antibiotics, as well as the cytotoxicity and effects on inflammation that were never addressed before.

## 2. Results

### 2.1. BrCl-flav Exhibits Potent Antibacterial Activity against ESKAPE Pathogens

The recorded MIC and MBC values showed that BrCl-flav poses important antibacterial activity against all clinical isolates used in this study (Table 1). The MICs for Gram-positive bacteria ranged between 0.24 µg/mL (recorded for two MRSA strains—*S. aureus* prxbio4 and *S. aureus* prxbio5) and 31.3 µg/mL (registered for *E. faecalis* prxbio8). The lowest MIC evidenced for Gram-negative bacteria was 0.48 µg/mL (*Haemophillus* spp. prxbio13), and the highest MIC value (125 µg/mL) was recorded for several strains—*K. pneumoniae*, *E. cloacae* and *S. enterica*. Regarding MBCs, lower values were also recorded for Gram-positive bacteria (0.48 µg/mL—MRSA strain) compared with Gram-negative bacteria (1.95 µg/mL—*Haemophillus* spp. prxbio13). Additionally, we need to emphasize that BrCl-flav showed more potent antibacterial activity (up to 16-fold higher) compared to chloramphenicol for several *S. aureus* strains, as well as for *S. pneumoniae* prxbio10 and *Haemophillus* spp. prxbio13. Comparable activity to gentamicin was recorded against *K. pneumoniae* medbio6-2013. However, for most of the Gram-negative isolates tested, BrCl-flav showed lower activity compared with gentamicin.

Based on antibiotic resistance profile and MIC/MBC values, several strains were selected to perform further tests using BrCl-flav as an antibacterial agent.

#### 2.1.1. BrCl-flav Induced a Concentration- and Time-Dependent Bacteriostatic Effect

The activity of BrCl-flav on the growth of selected clinical isolates over time was evaluated using concentrations equivalent to ½ × MIC, MIC and 2 × MIC (Figure 2). Compared to the control, BrCl-flav displayed dose-dependent and time-dependent bacteriostatic effects on all tested bacterial strains, as the growth curve analyses revealed. Thus, concentrations corresponding to ½ × MIC induced a significant growth delay (*p* ≤ 0.0056) up to 7 h (recorded for penicillin-resistant *S. aureus* prxbio1). Increasing the concentrations to the corresponding MIC values revealed a progressive inhibition of the growth of all tested strains up to 12 h compared with control. We must emphasize that no growth was detected by spectrophotometric measurements for *S. aureus* prxbio1 and *A. baumannii* medbio3-2013 cells exposed for more than 12 h to BrCl-flav at concentrations equivalent to 2 × MIC. Moreover, the analysis of the bacterial growth dynamics also revealed that the growth of the *S. aureus* prxbio1 strain was suppressed up to 24 h at 2 × MIC, showing that BrCl-flav has important bacteriostatic activity.

#### 2.1.2. BrCl-flav Possesses Important Bactericidal Activity against Selected Clinical Isolates

Time-killing curves of BrCl-flav were performed using the MBC values as reference. A total kill effect (no viable cells) was recorded for all selected strains at different exposure times (Figure 3). This effect was evidenced after only 30 min of incubation of *A. baumannii* medbio3-2013 cells in the presence of the tested antibacterial agent. We must emphasize that no viable cells were evidenced up to 24 h of exposure, showing the significant bactericidal potency of BrCl-flav against all tested resistant bacterial strains.

### 2.2. The Anti-Biofilm Activity of BrCl-flav

A biofilm formation assay was employed to determine the capability of the clinical isolates to form biofilms in vitro. The results are presented in Appendix A (Supplementary Files). By far, *Acinetobacter baumannii* medbio3-2013 exhibited the best biofilm-forming capacity, being classified as a strongly adherent strain. Based on this ability, the strain was selected as a representative ESKAPE pathogen for further tests regarding the anti-biofilm activity of BrCl-flav.

#### 2.2.1. Bacterial Biofilm Formation Was Inhibited by BrCl-flav

The formation of *A. baumannii* medbio3-2013 biofilms in the presence of BrCl-flav was significantly inhibited at different concentrations compared to the biofilms formed by unexposed cells (Figure 4a). BrCl-flav at concentrations between 3.9 and 62.5 μg/mL inhibited the biofilm formation more than 95% compared to the control. Lower and non-significant biofilm inhibition was recorded for ½ MIC (29.73%).

#### 2.2.2. BrCl-flav Showed Important Biofilm Disruption Potential

Because BrCl-flav exhibited encouraging inhibitory activity against biofilm formation, we proceeded with further experiments designed to assess the possible disruptive potential of the mature biofilms. Thus, *A. baumannii* medbio3-2013 biofilms were allowed to develop for 24 h prior to BrCl-flav exposure. Our data revealed that BrCl-flav displayed an important biofilm disruption potential (Figure 4b). Concentrations equivalent to MIC (3.9 μg/mL) and ½ MIC (1.95 μg/mL) induced a significant reduction of biofilm biomass of 57.71% and 39.91%, respectively.

### 2.3. Mode of Action

The mechanism of activity against representative Gram-positive and Gram-negative bacteria was investigated using fluorescence microscopy and SEM.

#### 2.3.1. BrCl-flav Impair the Cellular Membrane Integrity

Fluorescence microscopy was used to assess the uptake of PI fluorescent dye into *S. aureus* medbio1-2012 and *E. coli* medbio4-2013 cells with injured membranes after exposure to BrCl-flav at concentrations equivalent to MBC (Figure 5a,b). The low levels of fluorescence detected in control cells during the entire experiment confirmed the inability of PI to penetrate viable cells with intact membranes [25]. On the contrary, exposing *S. aureus* and *E. coli* cells to BrCl-flav significantly increased over time the number of fluorescent cells compared to control, most likely due to the increased permeability of the cellular membrane to PI. After only 25 min of incubation in the presence of the antibacterial agent, the percentage of *S. aureus* and *E. coli* fluorescent cells was 75.02% and 87.93%, respectively. A 100% fluorescent cell percentage was recorded after approximatively 1 h (*E. coli*) and 2 h (*S. aureus*) of BrCl-flav exposure (Figure 5).

As the fluorescence dynamics depicted in Figure 6 shows, the first fluorescent cells were detected after only 2 and 3 min of exposing *S. aureus* and *E. coli* cells, respectively, to BrCl-flav.

#### 2.3.2. The Effects of BrCl-flav on Bacterial Cell Morphology

SEM analysis revealed significant cell morphological damages of the BrCl-flav exposed cells. Control groups presented cells with normal morphologies (spheric coccus and short rods), clear boundaries and smooth surfaces (Figure 7). *S. aureus* medbio1-2012- and *E. coli* medbio4-2013-exposed cells were deformed, collapsed and had wrinkled surfaces (Figure 7). Moreover, the cell shrinkage was evidenced along with cellular debris, most likely resulting from the lysis of the cells, confirming the impairment of cellular membrane integrity.

### 2.4. Effect of BrCl-flav in Combination with Antibiotics against a MRSA Strain

FICIs were evaluated to investigate whether BrCl-flav used in combination with penicillin, ciprofloxacin and tetracycline provided synergistic or additive effects against an MRSA clinical isolate. The results showed additive responses in the case of BrCl-flav in combination with all three tested antibiotics (Table 2). Penicillin is the only antibiotic for which synergistic effects were recorded in combination with the tested antibacterial agent (FICI values: 0.5–0.264). When used in combination, the MICs of the two agents were reduced 68-fold for BrCl-flav and up to 4-fold for penicillin.

The antibacterial activity of one synergistic combination, BrCl-flav–penicillin (0.03/32 µg/mL), was further evaluated using a time-kill assay. No significant reduction of the viable cell number was recorded when BrCl-flav and penicillin were used alone, compared with the control (Figure 8). However, significant bactericidal activity was evidenced when the two agents were used in combination, with a total kill (no viable cells) effect recorded after 24 h (Figure 8).

### 2.5. BrCl-flav Effect on Human Cell Viability

Four cell lines were chosen for their different phenotypes (intestinal epithelial and epithelial mucous-producing cells: Caco-2 and HT29-MTX; hepatocytes: HepG2; and macrophages: U937) to evaluate the relative cytotoxicity of BrCl-flav by determining IC_50_ toward human cells (Figure 9).

The results obtained showed that the monocytes differentiated in macrophages appeared to be the most sensitive cell line to BrCl-flav (IC_50_ = 5.30 µg/mL), followed by hepatocytes (IC_50_ = 13.16 µg/mL). Epithelial cells appeared to be much more tolerant, as the calculated IC_50_ for the goblet-like cells (HT29-MTX) was 31.86 µg/mL. The IC_50_ calculation for Caco-2 was not possible regarding the increasing viability effect of BrCl-flav at concentrations ranging from 5 to 25 µg/mL.

### 2.6. Inflammation Study

To evaluate the pro- or anti-inflammatory effects of BrCl-flav, the secretion of a pro-inflammatory cytokine (TNF-α) and an anti-inflammatory cytokine (IL10) were quantified on LPS-induced macrophages. As shown in Figure 10, the secretion of the two cytokines studied was increased by exposure to LPS alone compared to the non-inflamed control.

The mean concentration of cytokines increased from 82.50 ± 165.00 pg/mL and 118.10 ± 3.06 pg/mL for the non-inflamed control to 2121 ± 281.7 pg/mL and 1707 ± 176.1 pg/mL for the LPS control, for TNF-α and IL10, respectively. The two cytokines, whether pro- or anti-inflammatory, were downregulated by the addition of glucocorticoid dexamethasone (positive control of inflammation inhibition) at the concentration of 20 µM. Indeed, TNF-α and IL10 concentrations decreased to 136.80 ± 23.55 pg/mL and 155.80 ± 30.00 pg/mL, corresponding to 95.55% and 90.87% inhibition, respectively.

BrCl-flav enhanced the secretion of TNF-α by LPS-stimulated macrophages by approximately 44% and decreased the IL10 anti-inflammatory cytokine secretion by 59%, both at all tested concentrations (not in a dose-dependent manner) (Figure 10).

## 3. Discussion

Antibiotic-resistant bacteria, particularly ESKAPE pathogens, are considered a global threat to human health. The acquisition of antimicrobial-resistance genes by ESKAPE pathogens has reduced the treatment options, increasing death rates due to treatment failure and stimulating the interest in the development of new antimicrobial therapies [26]. A possible solution could be the discovery of new molecules, not used until now in clinical therapy, for which the pathogens have not yet developed resistance. In this context, we hypothesize that BrCl-flav—a representative of a new class of synthetic sulfur containing tricyclic flavonoids with different halogen substituent at the benzopyran core—could be a reliable candidate for the formulation of new effective antimicrobials. We previously reported the important bactericidal and fungicidal effects of BrCl-flav [23,24,27]. However, no investigations were carried out on clinical bacterial isolates with different antibiotic resistance profiles. Additionally, no information is available on the cytotoxicity and anti-inflammatory activity. Here, we report the potent antibacterial activity against antibiotic-resistant bacteria from the ESKAPE group, low cytotoxicity and pro-inflammatory effect of BrCl-flav.

Determination of minimum inhibitory concentration revealed the important antibacterial activity of BrCl-flav against all bacterial strains tested in vitro. The most sensitive clinical isolates to BrCl-flav were MRSA strains, for which the lowest MICs values (0.24 µg/mL) were recorded. Moreover, our compound showed a more pronounced antibacterial activity against several *S. aureus*, *S. pneumoniae* and *Haemophillus* spp. strains compared to chloramphenicol, and comparable activity against one *K. pneumoniae* strain compared to gentamicin. Our data analysis also showed that Gram-negative bacteria were less susceptible to BrCl-flav, with the highest MIC of 125 µg/mL (registered for ESKAPE pathogens such as *K. pneumoniae* or *Enterobacter* spp.), compared to Gram-positive bacteria for which the highest MIC value recorded was 31.25 µg/mL (registered for an *E. faecalis* strain). This different susceptibility between Gram-positive and Gram-negative bacteria could be explained by the different cell wall structure and composition. Thus, the presence of the outer membrane in the Gram-negative cell wall provides protection against different antimicrobials, explaining the milder effect of BrCl-flav [28]. In this case, due to the strong ionization and the pronounced hydrophilic character, the tricyclic flavonoid presents a lower capability to penetrate the external hydrophobic membrane of the Gram-negative bacteria. 

The important antibacterial activity of BrCl-flav against pathogens from the ESKAPE group was also revealed by the literature survey carried out. Our compound was up to 416-fold more active against *S. aureus* and MRSA strains compared to previous reported natural flavonoids such as 2,4,4′-trihydroxy-6′-methoxy-chalcone (MIC = 100 µg/mL) [29], corylifol (MIC = 16 µg/mL) [30], or kuraridin (MIC = 8 µg/mL) [31]. We must emphasize that BrCl-flav exhibited antibacterial activity against clinically isolated enterococci comparable to Panduratin A (MICs of ≤2 μg/mL), considered to be one of the most potent natural flavonoids described before [32]. Compared to synthetic flavonoids such as different chalcone derivatives with MICs ranging from 0.5 to 63 µg/mL [33,34,35,36], BrCl-flav displayed stronger anti-MRSA potential, being up to 262-fold more active. BrCl-flav showed promising activity against *P. aeruginosa* with MIC values comparable to lophirones B and C (MIC = 100 μg/mL), even if other authors reported lower MICs for quinoline-based chalcone poly(CPA-co-AA)—MIC of 3.91 μg/mL [30]. Studies on synthetic flavonoids against *A. baumannii* received limited attention. However, the reported data revealed that BrCl-flav has a more important antibacterial effect (MIC of 3.9 μg/mL) compared to 2′-methoxy- and 4′-trifluoromethyl-substituted chalcones, with MIC values lying between 100 and 125 µg/mL [37]. Our compound showed a milder activity against *Enterobacter* spp. compared with previously tested flavonoids (MIC of 35 μg/mL) [30]. Regarding the effect against *K. pneumoniae*, BrCl-flav exhibited significantly lower activity (MIC of 125 μg/mL) compared to isobavachalcone or quinazoline and dithiocarbamate-based chalcone hybrids for which a MIC of 0.5–8 µg/mL was reported [38,39,40]. All these findings highlight that BrCl-flav is in fact a compound with potent antibacterial activity against ESKAPE pathogens, comparable to both flavonoids and antibiotics.

The antibacterial activity of BrCl-flav against representative ESKAPE pathogens (*S*. *aureus*, *E. faecium* and *A. baumannii*) was also assessed using growth kinetic studies. A significant decrease in the growth rate of all tested strains was detected when BrCl-flav was used at different concentrations, starting with the corresponding ½ × MIC. Exposing bacterial cells to concentrations equivalent to MIC induced a significant growth delay compared with control cells up to 12 h, as the growth curves of *S*. *aureus* prxbio1 show. When BrCl-flav was tested at concentrations corresponding to 2 × MIC, the growth of all tested pathogens was progressively inhibited, implying a significant dose-dependent inhibitory effect. MRSA and *A. baumannii* strains were most affected by BrCl-flav exposure at concentrations equivalent to 2 × MIC by diminishing the growth rate and reducing the final cellular density. We must highlight that no turbidity was revealed by the spectrophotometric measurements for all strains up to 12 h or up to 24 h for *S. aureus* prxbio1, denoting important bacteriostatic activity. 

After confirming that BrCl-flav possesses excellent antibacterial activity against multidrug-resistant clinical isolates, we evaluated further the bactericidal effect of BrCl-flav using the MBC assay. Our results showed that MBC values ranged between 0.48 and 250 µg/mL, with a MRSA strain being most susceptible. Compared to previously reported flavonoids such as taxifolin-7-O-α-L-rhamnopyranoside (MBC = 128 µg/mL) [41], luteolin (MBC = 2.5 μg/mL), 3′-O-methyldiplacol (MBC = 4 μg/mL), 2-(3,5-dihydroxy-4-methoxy-phenyl)-3,5-dihydroxy-8,8-dimethyl-2,3-dihydro-8*H*-pyrano[3,2]chromen-4-one (MBC = 50 μg/mL), sophoraflavanone (MBC = 1 μg/mL) [42] or (5-carbomethoxymethyl-4′,7-dihydroxyflavone (MBC = 50 μg/mL) [42], BrCl-flav expressed a more pronounced bactericidal activity against MRSA or *E. coli* strains. When the MBC/MIC ratio was calculated, bactericidal activity was evidenced against most of the strains (ratio of 2 or 4), except *S. aureus* prxbio6 (bacteriostatic activity was recorded), *K. pneumoniae*, *P. aeruginosa* and *S. enterica* (the calculation of the MBC/MIC ratio was not possible). To confirm the bactericidal potential of BrCl-flav a time-kill kinetics assay was employed using equivalent MBC as a reference. The complete killing of *A. baumannii*, *E. faecium* and *S. aureus* cells was observed after 30 min, 1 h and 3 h of exposure to BrCl-flav, suggesting a significant bactericidal effect. No viable cells were detected after 24 h of incubation in the presence of the tested compound, denoting a total kill effect. The results regarding the antibacterial activity are in accordance with our previous work on BrCl-flav; MICs of 0.24 and 3.9 μg/mL and MBCs of 0.24 and 7.8 μg/mL were recorded, respectively, for non-pathogenic *S. aureus* and *E. coli* strains [22,24].

Biofilm formation is related to the virulence potential of many bacterial strains, including ESKAPE pathogens. Infections caused by biofilm forming bacteria are very difficult to treat with current antibiotics; therefore, prevention of early-stage biofilm formation is essential for the treatment of these infections [13]. BrCl-flav showed important dose-dependent anti-biofilm activity, significantly inhibiting biofilm formation of *A. baumannii* with more than 95% at concentrations equivalent to the MIC. Because biofilms pose a serious medical challenge that is difficult to control, it is essential to find new agents that are able to eradicate biofilms [43]. Therefore, we examined the capability of BrCl-flav to disrupt mature biofilms of *A. baumannii*. Our compound showed important concentration-dependent biofilm-disruptive activity. At a low concentration of 1.95 μg/mL (corresponding to ½ MIC), BrCl-flav disrupted more than 39.91% of the biofilm mass and more than 50% at concentrations equivalent to 2 × MIC. The inhibition of biofilm formation by flavonoids was previously reported [42]. However, BrCl-flav showed higher anti-biofilm activity against *A. baumanni* compared with some natural flavonoids such as fisetin, phloretin and curcumin, reported to decrease biofilm formation of MDR *A. baumannii* strains with 46 and 93% at concentrations of 20 or 100 μg/mL [44]. Altogether, our results suggest that BrCl-flav possesses important anti-biofilm activity against *A. baumannii* and could represent a reliable solution to treat bacterial biofilm-dependent infections.

It has been shown so far that BrCl-flav has important antibacterial potential against ESKAPE pathogens. For the development of new therapeutic solutions, it is important to know the mechanism of action. We previously showed that BrCl-flav interferes with the cell membrane integrity of non-pathogenic bacterial strains [24]. Therefore, we investigated the effect of BrCl-flav on two multidrug-resistant *S. aureus* and *E. coli* strains using fluorescence microscopy. Cells with a damaged membrane considered to be dead or dying will appear stained red, while cells with an intact membrane will stain green when a Live/Dead BacLight bacterial viability kit is used. Exposing the Gram-positive and Gram-negative bacterial cells to concentrations of BrCl-flav equivalent to MBC revealed that the number of red fluorescent cells increased in a time-dependent manner. A percentage of 100% red fluorescent cells was reported for *E. coli* after approximatively 1 h and for *S. aureus* after 2 h of exposure to BrCl-flav, suggesting massive uptake of PI fluorescent dye and severe cell membrane damage. To verify the membrane type mechanism of action, SEM was employed to determine morphological damage induced by BrCl-flav exposure. Severe morphological alterations of *S. aureus* and *E. coli* cells were revealed by SEM analysis, together with cellular debris, sustaining the hypothesis that BrCl-flav targets the cellular membrane, inducing membrane structure alteration and cell lysis. Those effects could be a consequence of other mechanisms of action [19]. For instance, BrCl-flav could interact with purinic bases from bacterial DNA with the electrophilic C(2) atom of the 1,3-dithiolium ring after a Maxam–Gilbert mechanism, causing cell death followed by cell lysis [45]. However, our data support the hypothesis of a primary membrane-type mechanism of action. Thus, fluorescence dynamics tests showed that the *S. aureus* cell membrane is permeabilized for PI after only 2 min of exposure to BrCl-flav, while the fluorescent dye penetrates *E. coli* cell membranes within the first 3 min of exposure (for technical reasons it was not possible to obtain relevant pictures before 3 min). In addition, our previous investigations revealed that the antibacterial activity of BrCl-flav compared with the precursor 3-N,N-diethylaminodithiocarbamates flavanone is the consequence of the appearance of the third fused cycle, the 1,3-dithiolium ring [22]. 1,3-Dithiolium systems are well known for the reactivity of the C(2)-position towards nucleophiles [46,47]. Thus, the excellent antibacterial properties of BrCl-flav could be the result of the interaction between nucleophilic moieties of membrane constituents with the electrophilic C(2) atom of the 1,3-dithiolium ring.

The treatment of infections caused by bacteria that are resistant to multiple antibiotics (e.g., MRSA) is a real medical challenge, and very few therapeutic options are available. One option lies in the combination of antibiotics with new compounds to exploit potential synergistic effects [48]. Therefore, we used in our study a MRSA strain (*S. aureus* medbio1-2012) resistant to penicillin, ciprofloxacin and tetracycline. Combinations of those three antibiotics with BrCl-flav were tested to evidence possible synergistic effects. Our results showed that additive effects were recorded for all three tested antimicrobials. However, synergistic combinations were identified only for penicillin (FICI values: 0.5–0.264). To verify the synergistic effect, one combination, BrCl-flav–penicillin (0.03/32 µg/mL), was further used in a time-kill assay. The results revealed a significant bactericidal effect of penicillin in combination with BrCl-flav with no viable cells recorded after 24 h, while the two agents used separately showed no effects on *S. aureus* medbio1-2012 viability. These findings agree with previously reported data that showed synergistic effects of natural and synthetic flavonoids such as rutin, morin, quercetin, galangin, phenolic compound, substituted chalcones or pentacyclic triterpenoids with different antibiotics against *S. aureus* [49,50,51,52,53]. The synergistic effects could be explained by different cell structure targeted by the two antimicrobials used in our study; penicillin inhibits cell wall synthesis, while BrCl-flav induces membrane alterations, enhancing penicillin uptake. Our results suggest that BrCl-flav could be a potential solution to solve a serious problem caused by bacterial resistance to β-lactam antibiotics.

Our data showed up to this point that BrCl-flav could be a reliable alternative to develop effective drugs used to combat ESKAPE pathogens. Therefore, a cytotoxicity study was performed with hepatocytes, macrophages, epithelial and epithelial–mucus-producing cells to assess the effect of BrCl-flav on the cell phenotypes that would be in contact with the compound after being orally absorbed. Results revealed that macrophages and hepatocytes are more sensitive to BrCl-flav when compared to intestinal epithelial and goblet-like cells. Moreover, the MICs recorded for different tested bacterial strains such as MRSA, *Streptococcus* spp. prxbio9, *S. pneumoniae* prxbio10, *E. faecium* medbio2-2012, *E. coli* medbio4-2013 and *Haemophillus* spp. prxbio13 were relatively low compared to IC_50_ values registered for the tested cell lines. For Caco-2 cells, the compound exerted very low toxicity with an estimated IC_50_ value of around 80 µg/mL. Surprisingly, BrCl-flav increased cell viability at concentrations ranging from 5 to 25 µg/mL. This result is surprising but reproducible and may be due to BrCl-flav metabolization by the cells. Altogether, our findings suggest that BrCl-flav could be a reliable candidate for the formulation of new effective antimicrobials.

The effects of BrCl-flav (at non-cytotoxic concentrations of 0.1, 0.5 and 1 µg/mL) on inflammation were studied by measuring two cytokines, TNF-α and IL10, on U937 cells differentiated into macrophages and stimulated by LPS. TNF-α is a pro-inflammatory cytokine, necessary for host defense against infectious agents [54]. However, excessive inflammatory cytokine production results in tissue damage, toxicity and cell death. Pro-inflammatory cytokine synthesis by macrophages can also be modulated and inhibited by cytokines such as IL10 (anti-inflammatory cytokine) [55]. In our study, dexamethasone, a molecule belonging to the glucocorticoid family, was used as a positive control of inhibition inflammation. Indeed, glucocorticoids are considered anti-inflammatory and protective molecules due to their capacity to inhibit gene expression of pro-inflammatory cytokines and are widely used for the treatment of inflammation [56]. The results obtained showed that BrCl-flav exerts a pro-inflammatory effect due to the enhancement of TNF-α secretion and the reduction of IL10 production. Inflammation is a normal protective response to kill infectious agents. For example, Flynn et al. demonstrated that TNF-α plays an important role in host resistance to mycobacterial infection [57]. Moreover, aza-alkyl lysophospholipids, just like BrCl-flav, can induce TNF-α production and IL10 inhibition in peripheral blood-derived monocytes and therefore have a beneficial action in fighting microbial infections [58].

## 4. Materials and Methods

### 4.1. Antibacterial Agents

The synthesis of tricyclic flavonoid BrCl-flav has been described in detail in our previous report [59]. NMR, MS, IR and elemental analysis were used to determine the structure and purity (>99%) of the final compound. The stability of BrCl-flav towards Mueller–Hinton broth (Scharlau, Barcelona, Spain) and phosphate buffer saline (PBS) was monitored by UV–Vis spectroscopy. The tricyclic flavonoid was stable during the performed antimicrobial tests.

The antibiotics used in this study were purchased from local suppliers (Carl Roth and Sigma-Aldrich, Darmstadt, Germany, Scharlau, Barcelona, Spain); chloramphenicol and gentamicin were used as reference antibiotics for minimum inhibitory concentration assays, while ciprofloxacin, penicillin and tetracycline were used for combination tests; different antibiotics (Oxoid, Basingstoke, UK) were used for the antimicrobial susceptibility assay according to CLSI guidelines [60].

### 4.2. Bacterial Strains

*S. aureus* medbio1-2012, *S. aureus* prxbio1, *S. aureus* prxbio2, *S. aureus* prxbio3, *S. aureus* prxbio4, *S. aureus* prxbio5, *S. aureus* prxbio6, *S. aureus* prxbio7, *E. faecium* medbio2-2012, *E. faecalis* prxbio8, *Streptococcus* spp. prxbio9, *S. pneumoniae* prxbio10, *A. baumannii* medbio3-2013, *Escherichia coli* medbio4-2013, *E. cloacae* medbio5-2013, *K. pneumoniae* medbio6-2013, *K. pneumoniae* prxbio11, *K. pneumoniae* prxbio12, *P. aeruginosa* medbio7-2013, *Salmonella enterica* medbio8-2013 and *Haemophillus* spp. prxbio13 were provided by Praxis Clinical Laboratory (Iasi, Romania). All clinical isolates were obtained from biological specimens (nasal and conjunctival secretion, fecal, sputum, urine, pharyngeal and vaginal samples) and identified using the VITEK 2 Compact system (BioMérieux, Marcy l’Etoile, France). *S. aureus* ATCC 43300 and *K. pneumoniae* ATCC BAA-1705 were used as control strains. All strains were stored at −80 °C in 15% glycerol stocks. Before testing, the strains were transferred on Mueller–Hinton agar (MHA, Scharlau, Barcelona, Spain) and incubated at 37 °C for 24 h. Then, 10 mL of MHB was inoculated with one representative colony of bacteria taken from solid media, cultured at 37 °C for 24 h (190 rpm) and used as source of inoculum for each experiment.

### 4.3. Antibacterial Susceptibility Testing

#### 4.3.1. Disk Diffusion Method

The antibiotic resistance profile of the clinical isolates used in this study was determined by Kirby–Bauer’s disk diffusion method on Muller–Hinton agar, following the guidelines of Clinical and Laboratory Standards Institute [60]. The results are presented in Appendix A (Supplementary Files).

#### 4.3.2. Determination of Minimum Inhibitory Concentration and Minimum Bactericidal Concentration

Minimum inhibitory concentration (MIC) and minimum bactericidal concentration (MBC) were determined by the broth microdilution method as we previously described [24]. Briefly, BrCl-flav was serially diluted in MHB (a concentration range from 0.12 to 250 µg/mL) using 96-well plates and DMSO (Merck, Darmstadt, Germany) as a solvent. Inoculum, represented by bacterial suspensions with a cell density adjusted to approximately 2 × 10^6^ CFU/mL (CFU = colony-forming units) was added into each well of the microplate. DMSO at concentrations ranging from 0.006 to 12.5% (*v*/*v*), MHB medium and inoculum served as the control. MHB medium and inoculum were used as growth control. *S. aureus* ATCC 43300 and *K. pneumoniae* ATCC BAA-1705 were used as reference strains. Chloramphenicol and gentamicin were used as reference antibiotics. The lowest concentration with no visible growth after 24 h at 37 °C was considered as the MIC. To evaluate MBC, a volume of 15 µL taken from each well with no visible growth was inoculated on MHA plates. The MBC was considered as the lowest concentration at which bacteria failed to grow after plating onto MHA.

### 4.4. Growth Inhibition Assay

The effect of BrCl-flav on bacterial growth was assessed using the method described by Babii et al. [24], with some modifications. A volume of 250 µL from an overnight preculture was added in 25 mL MHB supplemented with BrCl-flav at final concentrations equivalent to ½ MIC, MIC and 2 × MIC (final cell density of approximately 10^6^ CFU/mL). Inoculated MHB medium supplemented with DMSO was used as a control. All flasks were incubated at 37 °C for 24 h under shaking conditions (190 rpm). The growth was monitored by measuring the optical density (OD) at 600 nm of samples taken at each hour up to 12 h and at 24 h, using a Beckman Coulter DU 730 spectrophotometer (Danaher Corporation, Washington, DC, USA).

### 4.5. Time-Kill Kinetic Assay

The killing rate of BrCl-flav was determined by measuring the reduction in the number of CFU per mL, following the procedure adapted after Aqil et al. [61]. The bacterial cells were harvested by centrifugation (4000 rpm, 20 min) and washed twice with PBS. Cell density was adjusted to obtain a bacterial suspension of approximately 10^8^ CFU/mL. A volume of 100 µL from this cell suspension served as inoculum and was added to 10 mL PBS supplemented with a concentration of BrCl-flav equivalent to MBC value (final cell density of approximately 10^6^ CFU/mL). The control was prepared similarly using DMSO at the appropriate concentration. All flasks were incubated at 37 °C for 24 h under shaking conditions (190 rpm). Samples were removed at each hour up to 12 h and at 24 h, serially diluted in PBS, plated onto MHA and incubated at 37 °C for 24 h. Afterwards, the CFU per mL was calculated by colony counting and transformed into log10 values. Bactericidal activity was defined as a ≥3 log10 reduction in the total CFU/mL from the original inoculum. Time–kill curves were constructed by plotting mean colony counts versus time [24].

### 4.6. In Vitro Anti-Biofilm Activity Assay

The clinical isolates ability to form biofilms was assessed using the method proposed by Onsare et al. [62] with some modifications. Static biofilm formation was evaluated in 96-well plates with lids (Becton Dickinson, Franklin Lakes, NJ, USA). A volume of 200 µL bacterial cells in MHB (approximately 10^6^ CFU/mL) was added into each well of a microtiter plate and cultured for 24 h at 37 °C. Uninoculated MHB medium served as control. The quantification of the bacterial biofilms was performed using a crystal violet assay [63]. A Beckman Coulter spectrophotometer was used to determine the ODs at a wavelength of 595 nm. To identify the strains with biofilm-forming ability, the following formula was used:$$\mathrm{ODc}=\mathrm{average}\;\mathrm{OD}\;\mathrm{of}\;\mathrm{negative}\;\mathrm{control}+\left(3\times \mathrm{standard}\;\mathrm{deviation}\;\mathrm{of}\;\mathrm{negative}\;\mathrm{control}\right)$$

The clinical isolates were classified as follows: OD ≤ ODc = non-adherent strains, ODc < OD ≤ 2 × ODc = weakly adherent strains, 2 × ODc < OD ≤ 4 × ODc = moderately adherent strains, 4 × ODc < OD = strongly adherent strains [64].

The same method as presented above was used to assess the anti-biofilm activity of BrCl-flav. For the inhibition of biofilm formation, MHB supplemented with different concentrations of antibacterial agent (samples) was inoculated with bacterial suspensions (final density adjusted to approximately 1 × 10^6^ CFU/mL). Inoculated MHB supplemented with DMSO served as a control. After incubation, the developed biofilm was assessed using crystal violet staining. For biofilm disruption assay, inoculated MHB was incubated for 24 h; following incubation, the culture medium was carefully discarded, and the wells were washed three times with PBS to remove non-attached cells. The same amount of MHB supplemented with different concentrations of BrCl-flav (samples) and DMSO (control) were added in each well, and the plate was incubated for another 24 h at 37 °C.

Biofilm formation as well as biofilm disruption in the presence of BrCl-flav were expressed as a percentage of the control biofilm formed in the absence of tested antimicrobial agent (considered as 100%), according to the following formula:$$Biofilm\;inhibition\;\left(\%\right)=\frac{\mathrm{OD}595\;\mathrm{nm}\;\mathrm{control}\;\mathrm{well}\;\mathrm{with}\;\mathrm{DMSO}-\;\mathrm{OD}595\;\mathrm{nm}\;\mathrm{experimental}\;\mathrm{well}\;\mathrm{with}\;\mathrm{BrCl}-\mathrm{flav}\;}{\mathrm{OD}595\;\mathrm{nm}\;\mathrm{control}\;\mathrm{well}\;\mathrm{with}\;\mathrm{DMSO}\;}$$

### 4.7. Evaluation of Cell Membrane Integrity

The integrity of cell membranes was assessed using fluorescence microscopy and the Live/Dead BacLight Bacterial Viability Kit (Invitrogen, Waltham, MA, USA), following the manufacturer’s instructions. Bacterial cells were harvested by centrifugation (4000 rpm, 20 min) and washed twice with PBS. Cell density was adjusted to approximately 10^8^ CFU/mL in PBS. A volume of 2 mL from the bacterial suspension was incubated in the presence of BrCl-flav at a concentration equivalent to the MBC value (125 µg/mL for *E. coli* medbio4-2013 and 62.5 µg/mL for *S. aureus* medbio1-2012) at 37 °C for 4 h under shaking conditions (190 rpm). Cells in PBS supplemented with DMSO served as control. Samples was taken at 30 min, 1, 1.5, 2, 3 and 4 h, washed twice with PBS and stained with SYTO 9 and propidium iodide (PI) for 15 min in the dark. The fluorescent cells were counted using a DM100 LED fluorescence microscope (Leica, Solms, Germany) and an I3 blue excitation range filter cube (BP 450 ± 490 nm band-pass filter). At least five random, independent images were captured per sample, and the ratio between green/red fluorescent cells and total cells was calculated as percentage. The dynamics of dye penetration into cells exposed to BrCl-flav at concentration equivalent to MBC was performed following the same protocol. Photographs were taken every minute, up to 10 min using a DM100 LED fluorescence microscope.

### 4.8. Scanning Electron Microscopy

Suspensions in PBS of the logarithmic growth phase of *S. aureus* medbio1-2012 and *E. coli* medbio4-2013 cells (approximately 1 × 10^8^ CFU/mL) were incubated in PBS for 6 h in the presence of BrCl-flav (final concentrations equivalent to MBC values). DMSO served as a control. Samples (untreated and treated bacterial cells) were prepared for SEM analysis following the protocol previously described [23] and examined with a Tescan Vega II SBH microscope using the secondary electron detector at an acceleration voltage of 30 kV.

### 4.9. Checkerboard Assay

The effect of BrCl-flav in combination with different antibiotics against *S. aureus* methicillin-resistant strain medbio1-2012 was assessed using the checkerboard microdilution method [65], with some modifications. Briefly, the synthetic flavonoid and the antibiotics were serial two-fold diluted in Eppendorf microtubes containing MHB medium. The concentrations of the antibiotics and BrCl-flav were selected based on previously determined MIC values. A volume of 50 µL of each compound dilution was added to each well of a microplate to obtain antibiotics and BrCl-flav concentration ranges of 0.125–256 µg/mL (penicillin), 1.22–156.25 µg/mL (ciprofloxacin), 0.15–19.53 µg/mL (tetracyclin) and 0.0002–0.96 µg/mL (BrCl-flav). Next, 100 µL of bacterial suspension was added to each well to reach a final cell density of approximatively 10^6^ CFU/mL. Inoculated MHB medium was used as a control. After 24 h of incubation at 37 °C, the bacterial growth was assessed visually in the presence of resazurin (0.05%) (Difco, Tucker, GA, USA). The MIC of the combined antibacterial agents was considered the lowest concentration at which no visible growth was observed.

To evaluate the combination effect, the fractional inhibitory concentration index (FICI) was calculated using the formula:FICI_AB_ = FIC_A_ + FIC_B_

The FICI results for each combination were defined as synergy (≤0.5), additivity (0.5 < FICI ≤1), indifference (1 < FICI ≤4) and antagonism (>4) [49]. Additionally, the effect of a selected synergistic combination was further assessed by time–kill curves, as presented above.

### 4.10. Cell Culture

Human colonic epithelial cells, Caco-2 (accession No. 86010202, lot 16H030, ECACC), human Caucasian colon adenocarcinoma grade II cell line HT29-MTX-E12 (accession N° 12040401, lot 16D017, ECACC) and human Caucasian hepatocyte carcinoma cells HepG2 (accession No. 85011430, lot 16K046, ECACC) were cultured in monolayers in Dulbecco’s Modified Eagle Medium (DMEM) substituted with 10% heat-inactivated fetal bovine serum (FBS) (Gibco Laboratories, Grand Island, NY, USA), 100 U/mL penicillin/0.1 mg/mL streptomycin (PanBiotech GmbH, Hamburg, Germany) and 2 mM L-glutamine (PanBiotech GmbH, Germany). Cells were grown at 37 °C and 5% CO_2_ (CO_2_ Incubator, Thermo Scientific, Marietta, GA, USA) and sub-cultured when the monolayers reached 90% confluence. For sub-culturing, the monolayers were treated with trypsin–EDTA (PanBiotech GmbH, Germany).

The human Caucasian histiocytic lymphoma cell line U937 (accession No. 85011440, lot No. 11D008) was cultured in Roswell Park Memorial Institute (RPMI) 1640 Medium supplemented with heat-inactivated FBS (10%) penicillin (100 U/mL)/streptomycin (0.1 mg/mL) and L-glutamine (2 mM). Cells were grown at 37 °C and 5% CO_2_ and passaged every 2–3 days.

### 4.11. U937 Differentiation into Macrophages

U937 cells were differentiated into macrophages with 25 nM phorbol 12-myristate 13-acetate (PMA, Sigma-Aldrich, St. Louis, MO, USA) for 48 h, followed by a three-day incubation without PMA.

### 4.12. Cell Viability CCK-8 Assay

Caco-2 and HT29-MTX cells were seeded at 5.8 × 10^3^ cells/well, HepG2 cells were seeded at 2 × 10^4^ cells/well, and U937 cells differentiated into macrophages were seeded at 1 × 10^5^ cells/well in 96-well culture plates. Cells were incubated with BrCl-flav at increasing concentrations (0.1 to 100 µg/mL) for 24 h. Cell viability was determined by cell counting kit-8 (CCK-8) assay (Dojindo Laboratories, Kumamoto, Japan) according to the manufacturer’s protocol. Briefly, 2-(2-methoxy-4-nitrophenyl)-3-(4-nitrophenyl)-5-(2,4-disulfophenyl)-2H-tetrazolium, monosodium salt (WST-8) was added to cells after the incubation with samples, at the final concentration of 5%. Then the cells were incubated at 37 °C for 90 min in the dark. WST-8 was reduced at the intracellular level to give an orange-colored product (formazan). The amount of formazan generated was directly proportional to the number of viable cells. Absorbance was measured at 450 nm using a microplate spectrophotometer (SpectraMax^®^ iD3, Molecular Devices, San Jose, CA, USA).

### 4.13. Inflammation Study

The purpose of this study was to investigate the effects of BrCl-flav on the inflammatory responses to lipopolysaccharide (LPS)-induced U937 macrophage cells. For this, U937 cells were differentiated into macrophages as previously described, plated into 12-well plates at a density of 2 × 10^6^ viable cells/well and incubated with LPS from *E. coli* O26:B6 (L2654, Sigma-Aldrich, St. Louis, MO, USA) at 50 μg/mL and BrCl-flav at the concentrations 0.1, 0.5 and 1 µg/mL for 4 h. RPMI was used as a negative control for inflammation, LPS alone as an inflammation control and 50 μg/mL-LPS/20 µM-dexamethasone as a positive inflammation inhibition control. The cell culture supernatants were then collected for further analysis of the secreted cytokines and stored at −20 °C. The levels of TNF-α and IL10 in macrophage culture media were measured by commercially available enzyme-linked immunosorbent assay (ELISA) kits according to the manufacturer’s instructions (R&D Systems, Minneapolis, MN, USA), by comparing the obtained optical densities (microplate spectrophotometer SpectraMax^®^ iD3) to the standard curve.

### 4.14. Statistical Analysis

Experiments were performed in triplicate. The statistical evaluation of the results was carried out by Dunnett’s multiple comparisons test, the data being presented as mean (n = 3) ± SEM. For the inflammation study, a Mann–Whitney test to compare the LPS control to the dexamethasone control and sample was used, and data are presented as mean (n = 4) ± SD. All data were analyzed using GraphPad Prim 9 software (GraphPad Software, Inc., La Jolla, CA, USA). Differences between groups were considered significant when *p* < 0.05.

## 5. Conclusions

BrCl-flav showed important in vitro inhibitory activity against antibiotic-resistant “priority pathogens” such as *S. aureus*, *A. baumannii*, *P. aeruginosa* and *E. faecium*. Additionally, the synthetic flavonoid expressed strong bactericidal activity with total kill in a very short time. Our compound inhibited biofilm formation and displayed important biofilm disruption potential against *A. baumannii*. Those effects are induced most likely by membrane integrity damage and cell lysis. BrCl-flav expressed synergistic antibacterial activity in combination with penicillin against an MRSA clinical isolate. Additionally, BrCl-flav showed low cytotoxicity and pro-inflammatory effects. These very promising results suggests that BrCl-flav is in fact a compound with potent antibacterial activity against representative ESKAPE pathogens, which may be used to develop new effective antimicrobial agents able to bypass bacterial multidrug resistance.

## Figures and Tables

**Figure 1 antibiotics-11-01389-f001:**
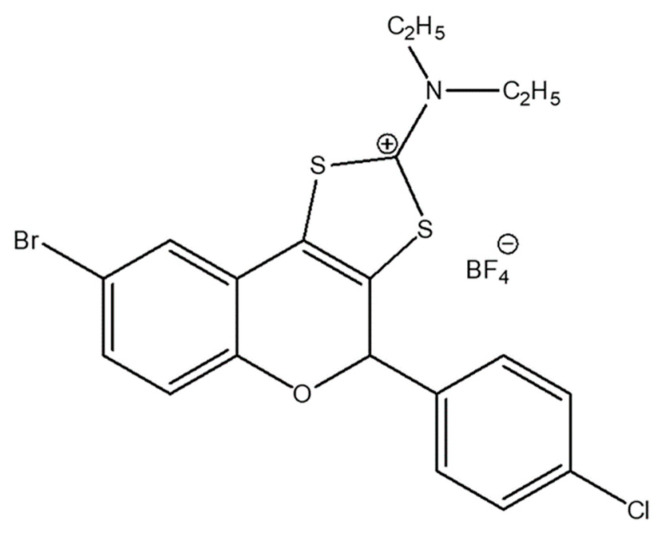
Structure of flavonoid BrCl-flav.

**Figure 2 antibiotics-11-01389-f002:**
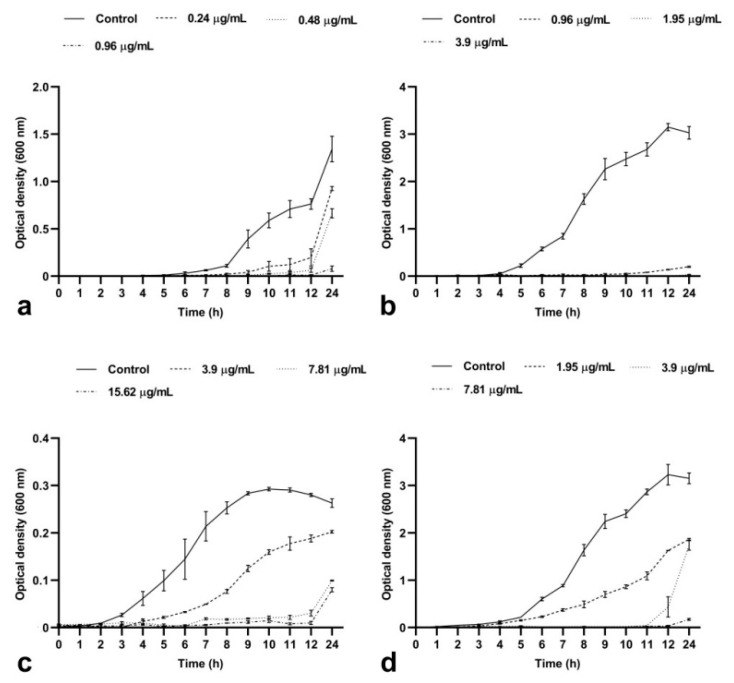
The effects of BrCl-flav at different concentrations on the growth of *S. aureus* medbio1-2012 (**a**), *S. aureus* prxbio1 (**b**), *E. faecium* medbio2-2012 (**c**) and *A. baumannii* medbio3-2013 (**d**). Cells incubated in MHB medium supplemented with DMSO served as control. The data were presented as mean of three independent experiments. Bars indicate SEM.

**Figure 3 antibiotics-11-01389-f003:**
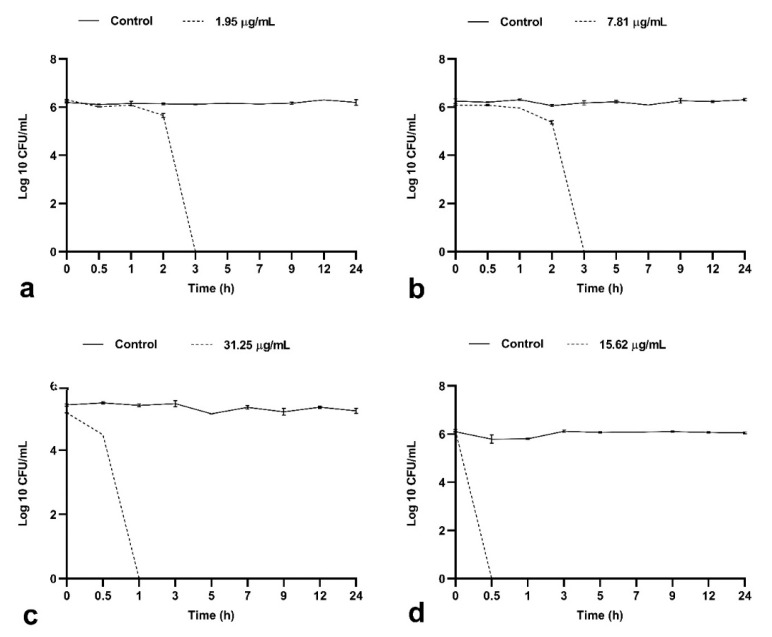
The time-killing kinetics of BrCl-flav against *S. aureus* medbio1-2012 (**a**), *S. aureus* prxbio1 (**b**), *E. faecium* medbio2-2012 (**c**) and *A. baumannii* medbio3-2013 (**d**). The bacterial cells were exposed to BrCl-flav at concentrations equivalent to MBC. Untreated cells served as control. The data were presented as the means of three independent experiments. Bars indicate SEM.

**Figure 4 antibiotics-11-01389-f004:**
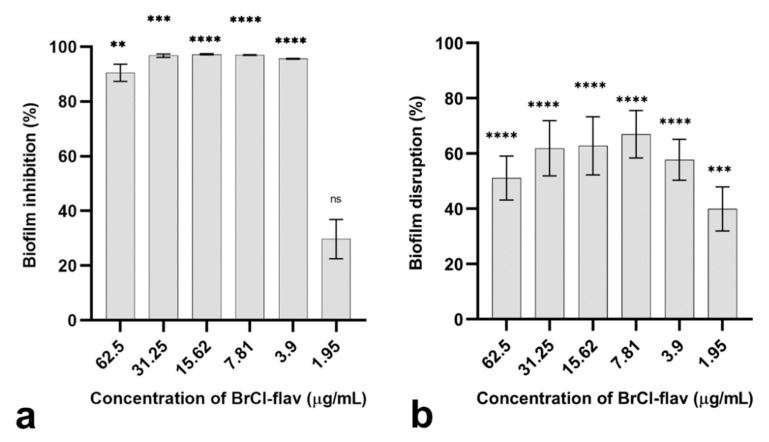
The effects of the BrCl-flav on *A. baumannii* medbio3-2013 biofilms: (**a**) inhibition of biofilm formation (bacterial cells were incubated for 24 h in the presence of BrCl-flav); (**b**) disruption of mature biofilms (before exposure to BrCl-flav, the biofilms were pre-formed for 24 h). Values are the means of at least three replicates. Bars indicate SEM. Asterisks represent a significant difference (*p* < 0.05) vs. Control (** = *p* < 0.01, *** = *p* < 0.001, **** = *p* < 0.0001; ns—non-significant).

**Figure 5 antibiotics-11-01389-f005:**
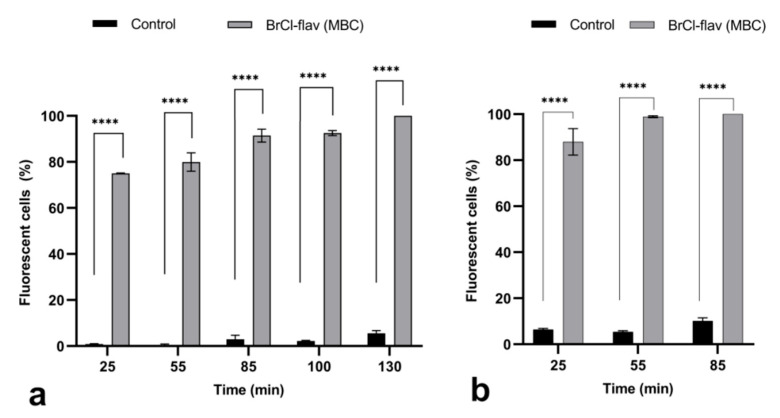
Effect of BrCl-flav exposure on *S. aureus* medbio1-2012 (**a**) and *E. coli* medbio4-2013 (**b**) cell membrane integrity. Exponential-phase cells were treated with the antibacterial agent concentration equivalent to MBC and stained with SYTO 9 and propidium iodide. Low levels of fluorescent cells were detected in the untreated control. Red fluorescent cells were detected in samples starting with 25 min of incubation with BrCl-flav, indicating membrane damages. Values are the means of three replicates. Bars indicate SEM. Asterisks denote a significant difference (*p* < 0.05) vs. Control (**** = *p* < 0.0001).

**Figure 6 antibiotics-11-01389-f006:**
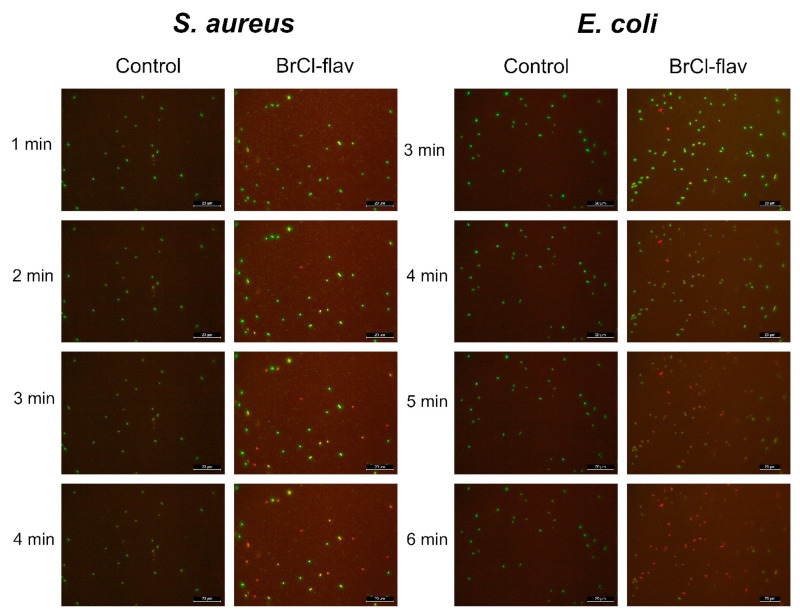
Fluorescent images of a time course experiment illustrating the *S. aureus* medbio1-2012 and *E. coli* medbio4-2013 cell membrane disruption effects due to BrCl-flav exposure, visualized by the influx of the fluorescent nuclear stain propidium iodide. The cells were stained using a Live/Dead BacLight Bacterial Viability Kit (Invitrogen, Waltham, MA, USA). The captured images are representative of a typical result.

**Figure 7 antibiotics-11-01389-f007:**
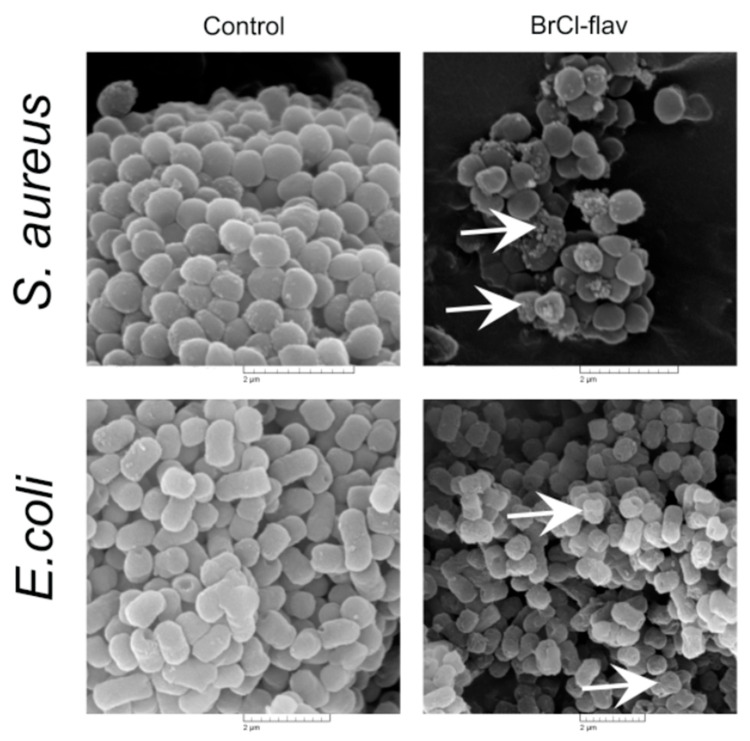
The effects of BrCl-flav on the morphology of *S. aureus* medbio1-2012 and *E. coli* medbio4-2013. Control cells with normal morphology. Cells exposed for 6 h to concentrations equivalent to MBC. White arrows indicate morphological damages and cellular debris.

**Figure 8 antibiotics-11-01389-f008:**
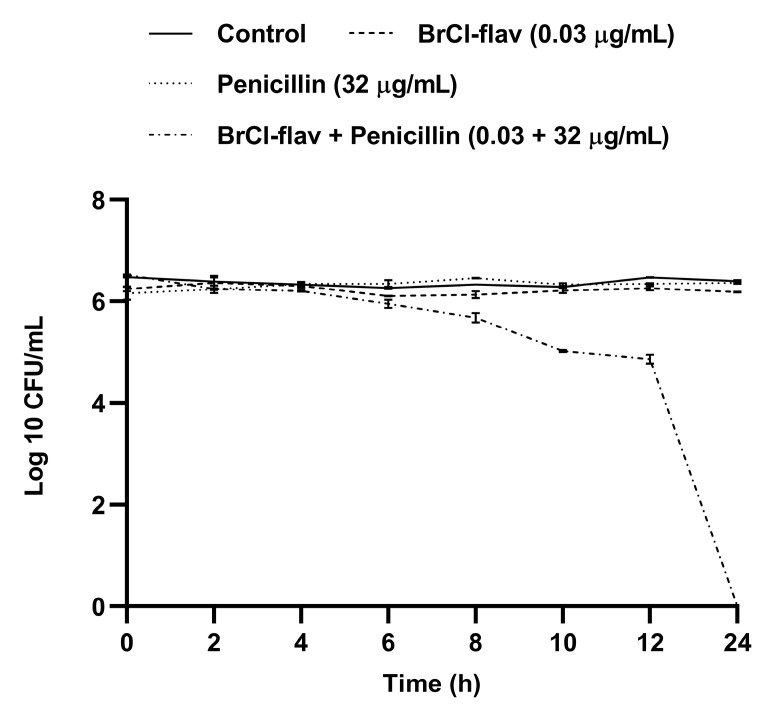
The time-kill curve of BrCl-flav and penicillin synergistic combination against an MRSA isolate. Values are the means of three replicates. Bars indicate SEM.

**Figure 9 antibiotics-11-01389-f009:**
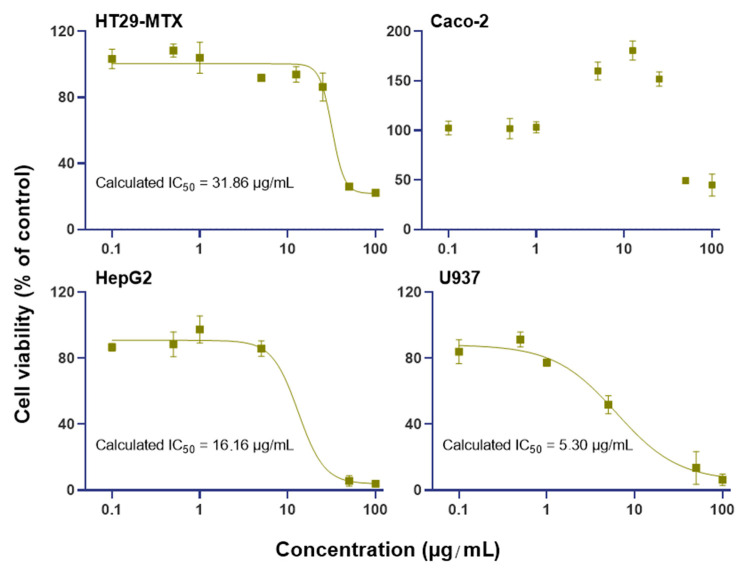
Effect of BrCl-flav on the viability of four human cell lines. Cells were incubated for 24 h with the BrCl-flav at increasing concentrations (0.1 to 100 µg/mL). The viability of the cells was evaluated by the measurement of mitochondrial hydrogenase activity assayed with CCK8 reagent. Means are presented ± SD (N = 2, n = 6). The molecule concentration required to cause 50% inhibition of the cell viability (IC_50_) was determined using the nonlinear regression analysis function of GraphPad Prism.

**Figure 10 antibiotics-11-01389-f010:**
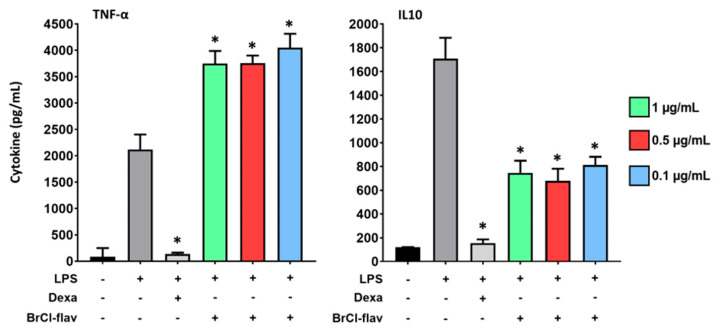
Effect of BrCl-flav on LPS-induced secretion of tumor necrosis factor-alpha (TNF-α) and interleukin 10 (IL10) in U937-macrophages. U937 cells were differentiated into macrophages by incubation in the presence of 25 nM-PMA for 48 h, and then after 3 days of PMA weaning, cells were seeded in 12-well plates at the density of 2 × 10^6^ viable cells/well. Cytokine production was measured after 4 h of incubation with culture medium (black), LPS at 50 μg/mL (dark grey), or with a positive inflammation inhibition control (LPS at 50 μg/mL + dexamethasone, 20 µM; light grey) or with LPS (50 μg/mL) + different concentrations of BrCl-flav (n = 4). Statistical analysis of the data was performed using Mann–Whitney tests to compare the LPS control to the dexamethasone control and samples with GraphPad Prism software (* = *p* < 0.05). Results are presented as bar graphs (means and standard deviations).

**Table 1 antibiotics-11-01389-t001:** Minimum inhibitory concentration and minimum bactericidal concentration of BrCl-flav against tested clinical isolates (µg/mL).

Bacterial Strains	BrCl-flav		DMSO	Control	
	MIC	MBC	MIC	MIC	MBC
*Staphylococcus aureus* medbio1-2012	0.48	1.95	>125	7.8 ^c^	31.25 ^c^
*S. aureus* prxbio1	1.95	7.81	>125	3.9 ^c^	125 ^c^
*S. aureus* prxbio2	7.81	15.6	250	7.8 ^c^	125 ^c^
*S. aureus* prxbio3	0.48	0.97	250	1.9 ^c^	62.5 ^c^
*S. aureus* prxbio4	0.24	0.97	250	3.9 ^c^	15.62 ^c^
*S. aureus* prxbio5	0.24	0.48	250	7.8 ^c^	62.5 ^c^
*S. aureus* prxbio6	7.81	250	250	7.8 ^c^	125 ^c^
*S. aureus* prxbio7	0.48	0.97	250	7.8 ^c^	62.5 ^c^
*Streptococcus* spp. prxbio9	3.9	7.81	250	<0.9 ^c^	>125 ^c^
*S. pneumoniae* prxbio10	0.48	0.97	250	3.9 ^g^	31.25 ^g^
*Enterococcus faecium* medbio2-2012	7.81	31.3	>125	<0.9 ^c^	15.62 ^c^
*E. faecalis* prxbio8	31.3	62.5	125	7.8 ^c^	> 125 ^c^
*Acinetobacter baumannii* medbio3-2013	3.9	15.6	>125	<0.9 ^g^	<0.9 ^g^
*Escherichia coli* medbio4-2013	3.9	15.6	>125	<0.9 ^g^	<0.9 ^g^
*Enterobacter cloacae* medbio5-2013	125	250	125	<0.9 ^g^	<0.9 ^g^
*Klebsiella pneumoniae* medbio6-2013	125	>250	125	>125 ^g^	>125 ^g^
*K. pneumoniae* prxbio11	125	>250	125	<0.9 ^g^	<0.9 ^g^
*K. pneumoniae* prxbio12	125	>250	125	<0.9 ^g^	<0.9 ^g^
*Pseudomonas aeruginosa* medbio7-2013	31.3	-	>125	<0.9 ^g^	<0.9 ^g^
*Salmonella enterica* medbio8-2013	125	-	>125	<0.9 ^g^	<0.9 ^g^
*Haemophillus* spp. prxbio13	0.48	1.95	250	0.9 ^g^	125 ^g^
*S. aureus* ATCC 43300	3.9	31.3	125	7.8 ^c^	62.5 ^c^
*K. pneumoniae* ATCC BAA-1705	125	250	125	3.9 ^g^	15.62 ^g^

MIC = minimum inhibitory concentration; MBC = minimum bactericidal concentration; ^c^—chloramphenicol; ^g^—gentamicin; the values are the means of at least three replicates.

**Table 2 antibiotics-11-01389-t002:** Fractional inhibitory concentration indices (FICIs) of BrCl-flav and antibiotics against a clinical isolate of MRSA.

Bacterial Strain	MIC (µg/mL)				FICI	Effect
	Alone		In Combination			
	BrCl-flav	Penicillin	BrCl-flav	Penicillin		
*S. aureus* medbio1-2012	0.48	128	0.96	8	2.06	IND
			0.48	16	1.125	IND
			0.24	32	0.75	ADD
			0.12	32	0.5	SYN
			0.06	32	0.375	SYN
			0.03	32	0.312	SYN
			0.015	32	0.281	SYN
			0.007	32	0.264	SYN
	**BrCl-flav**	**Ciprofloxacin**	**BrCl-flav**	**Ciprofloxacin**	**FICI**	**Effect**
	0.48	156.25	0.48	9.76	1.06	IND
			0.48	4.88	1.03	IND
			0.48	2.44	1.015	IND
			0.48	1.22	1.007	IND
			0.24	39.06	0.74	ADD
			0.24	19.53	0.62	ADD
			0.12	78.12	0.74	ADD
			0.06	78.12	0.615	ADD
			0.06	156.25	1.125	IND
			0.03	156.25	1.06	IND
			0.015	156.25	1.031	IND
			0.007	156.25	1.015	IND
			0.003	156.25	1.007	IND
			0.001	156.25	1.003	IND
			0.0009	156.25	1.001	IND
			0.0004	156.25	1.0009	IND
			0.0002	156.25	1.0004	IND
	**BrCl-flav**	**Tetracycline**	**BrCl-flav**	**Tetracycline**	**FICI**	**Effect**
	0.48	19.531	0.48	4.882	1.025	IND
			0.48	2.441	1.012	IND
			0.48	1.222	1.06	IND
			0.48	0.61	1.031	IND
			0.48	0.305	1.015	IND
			0.48	0.152	1.007	IND
			0.24	9.76	0.999	ADD
			0.12	19.531	1.25	IND
			0.06	19.531	1.125	IND
			0.03	19.531	1.06	IND
			0.015	19.531	1.031	IND
			0.07	19.531	1.015	IND
			0.003	19.531	1.007	IND
			0.001	19.531	1.003	IND
			0.0009	19.531	1.001	IND
			0.0004	19.531	1.0009	IND
			0.0002	19.531	1.0004	IND

MIC: minimum inhibitory concentration; SYN: synergy (FICI ≤ 0.5); ADD: additivity (0.5 < FICI ≤ 1); IND: indifference (1 < FICI ≤ 4); the values are the means of at least three replicates.

## Data Availability

Data are contained in the article.

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
