# Peer review of "Synthetic Flavonoid BrCl-Flav—An Alternative Solution to Combat ESKAPE Pathogens"

_antibiotics, 2022, doi:10.3390/antibiotics11101389_

Round 1
Reviewer 1 Report
Comments:
The article entitled “Synthetic flavonoid BrCl-flav – an alternative solution to com-2 bat ESKAPE pathogens” describes the potent in vitro antibacterial activity of BrCl- tricyclic flavonoids, a representative of a new class of synthetic tricyclic flavonoids ESKAPE pathogens. The article also describes the minimum inhibitory/bactericidal concentration, time kill and biofilm formation assays to estimate the antibacterial potential of BrCl-flavonoids. Further fluorescence and scanning electron microscopy were utilized to study the mechanism of action was. Checkerboard assay was used to study the effect of the tested compound in combination with antibiotics.
1. However, there are certain suggestions in this manuscript in which improvement is needed. The structure of the BrCl- tricyclic flavonoids, if represented initially in the manuscript, after the description or at the end of the introduction would give better idea for the readers about the structure.
2. Please include the following references in the introduction section.
1) “Clinical relevance of the ESKAPE pathogens” Expert Review of Anti-infective Therapy, Volume 11, 2013 - Issue 3, Page 297-308.
2) “Considerations and Caveats in Combating ESKAPE Pathogens against Nosocomial Infections” Advance Science, 2020, 7, 1901872
Decision: Accept after the above minor revision.
Author Response
Response to Reviewer 1 Comments
First, the authors would like to thank the reviewer for the patient and careful evaluation of our work and for providing ideas and corrections that will improve the quality of the manuscript.
Point 1: However, there are certain suggestions in this manuscript in which improvement is needed. The structure of the BrCl- tricyclic flavonoids, if represented initially in the manuscript, after the description or at the end of the introduction would give better idea for the readers about the structure.
Response 1: Following the reviewer suggestion, Figure 10 was moved at the end of Introduction section and became Figure 1. Consequently, all figures were renumbered.
Point 2: 2. Please include the following references in the introduction section.
1) “Clinical relevance of the ESKAPE pathogens” Expert Review of Anti-infective Therapy, Volume 11, 2013 - Issue 3, Page 297-308.
2) “Considerations and Caveats in Combating ESKAPE Pathogens against Nosocomial Infections” Advance Science, 2020, 7, 1901872.
Response 2: We thank the reviewer for this suggestion. Both references were included in the Introduction section.
Reviewer 2 Report
The article describes antibacterial activity of BrCl-flav against clinical bacterial isolates, as well as exploring mechanisms involved for the stated antibacterial activity.
Results
1. Page 2 Line 90: “…between 0.24-31.25µg/mL” - Please clarify which organism has got MIC of 31.25µg/mL after BrCl-flav treatment.
2. Figure 1 – Untreated cells or cells treated with vehicle was used as control?
3. Page 11 Line 274 – sensible or sensitive?
Discussion
1. Page 12 Line 321 – Please correct the phrase “bacterial clinical isolates” to “clinical bacterial isolates”.
2. Page 14 Line 433 – “…these is unlikely…” Please check on this sentence.
General comments
1. Could the authors perform MBC/MIC ratio for determination of whether it is bactericidal or bacteriostatic activity as well?
2. Did the authors check on the potential of drug resistant after treated with BrCl-flav?
Author Response
Response to Reviewer 2 Comments
First, the authors would like to thank the reviewer for the patient and careful evaluation of our work and for providing ideas and corrections that will improve the quality of the manuscript.
Point 1: Page 2 Line 90: “…between 0.24-31.25µg/mL” - Please clarify which organism has got MIC of 31.25µg/mL after BrCl-flav treatment.
Response 1: We thank the reviewer for his suggestion. Next sentence was modified in the manuscript:
The MICs ranged for Gram-positive bacteria between 0.24 µg/mL (recorded for two MRSA strains - S. aureus prxbio4 and S. aureus prxbio5) and 31.3 µg/mL (registered for E. faecalis prx8).
Point 2: Figure 1 – Untreated cells or cells treated with vehicle was used as control?
Response 2: As we stated in the M&M section (4.4.) cells cultivated in MHB medium supplemented with DMSO served as control. To avoid confusion, the capture of Figure 1 (new Figure 2) was modified as follows:
Figure 2 – The effects of BrCl-flav at different concentrations on the growth of S. aureus medbio1 – 2012 (a), S. aureus prxbio1 (b), E. faecium medbio2-2012 (c) and A. baumannii medbio3-2013 (d). Cells incubated in MHB medium supplemented with DMSO served as control. The data were presented as mean of three independent experiments. Bars indicate SEM.
Point 3: Page 11 Line 274 – sensible or sensitive?
Response 3: We thank the reviewer for spotting the mistake. Next change was performed in the manuscript:
The results obtained showed that the monocytes differentiated in macrophages appeared to be the most sensitive cell line to BrCl-flav (IC50 = 5.30 µg/mL), followed by hepatocytes (IC50 = 13.16 µg/mL).
Point 4: Page 12 Line 321 – Please correct the phrase “bacterial clinical isolates” to “clinical bacterial isolates”.
Response 4: The suggested correction was made in the manuscript.
Point 5: Page 14 Line 433 – “…these is unlikely…” Please check on this sentence.
Response 5: The sentence was revised as follows:
However, our data support the hypothesis of a primary membrane-type mechanism of action. Thus, fluorescence dynamics tests showed that S. aureus cell membrane is permeabilized for PI after only 2 min of exposure to BrCl-flav, while the fluorescent dye penetrates E. coli cell membranes within first 3 min of exposure (from technical reasons it was not possible to obtain relevant pictures before 3 min).
Point 6: Could the authors perform MBC/MIC ratio for determination of whether it is bactericidal or bacteriostatic activity as well?
Response 6: Thank you for this kind advice. We performed MBC/MIC ratio and we evidenced a bactericidal activity for the majority of the strains (ratio of 2 or 4), except S. aureus prxbio6 (bacteriostatic activity), K. pneumoniae and P. aeruginosa and S. enterica (the calculation of MBC/MIC ratio was not possible). Consequently, next phrase was introduced in the manuscript:
When the MBC/MIC ratio was calculated, a bactericidal activity was evidenced against most of the strains (ratio of 2 or 4), except S. aureus prxbio6 (a bacteriostatic activity was recorded), K. pneumoniae and P. aeruginosa and S. enterica (the calculation of MBC/MIC ratio was not possible).
Point 7: Did the authors check on the potential of drug resistant after treated with BrCl-flav?
Response 7: We thank the reviewer for this suggestion. We discussed also to check for resistance acquired after exposure to BrCl-flav and it will be a part of a future investigation.
Reviewer 3 Report
The authors explained the antimicrobial characterization BrCl-flav, already been reported before by them. The compound has potent antimicrobial properties but also has significant toxicity, which limits its potential use as an antibiotic. I will suggest accepting this manuscript for publication in Antibiotics after the following important changes.
1. The manuscript has a number of typos (see the attached file).
2. Authors have discussed extensively ESKAPE pathogens in the manuscript. I strongly suggest adding SPEAKS pathogens in the introduction and cite these two papers.
https://www.mdpi.com/2079-6382/11/7/939/htm
Antibiotics 2022, 11(7), 939; https://doi.org/10.3390/antibiotics11070939
https://www.thelancet.com/journals/lancet/article/PIIS0140-6736(21)02724-0/fulltext
3. Figure 10 should be n the introduction.
4. Authors are claiming this in the conclusion. "These very promising results suggests that BrCl-flav is in fact a compound 745 with a potent antibacterial activity against ESKAPE pathogens, which may be used to develop new effective antimicrobial agents able to bypass bacterial multidrug resistance."
I don't see antimicrobial data against all six bacteria of ESKAPE pathogens. So, the above statement is not true.

Author Response
Response to Reviewer 3 Comments
First, the authors would like to thank the reviewer for the patient and careful evaluation of our work and for providing ideas and corrections that will improve the quality of the manuscript.
Point 1: The manuscript has a number of typos (see the attached file).
Response 1: We thank the reviewer for spotting all those typos. All the corrections were performed in the revised manuscript.
Point 2: Authors have discussed extensively ESKAPE pathogens in the manuscript. I strongly suggest adding SPEAKS pathogens in the introduction and cite these two papers.
https://www.mdpi.com/2079-6382/11/7/939/htm
Antibiotics 2022, 11(7), 939; https://doi.org/10.3390/antibiotics11070939
https://www.thelancet.com/journals/lancet/article/PIIS0140-6736(21)02724-0/fulltext
Response 2: We thank the reviewer for his suggestion.
We introduced first recommended reference. The second one was already cited in the manuscript (former reference nr. 6). In addition, next sentence was introduced in the manuscript:
On the other hand, SPEAKS pathogens (S. aureus, P. aeruginosa, Escherichia coli, A. baumannii, K. pneumoniae, and S. pneumoniae) are the leading causes of AMR deaths
Point 3: Figure 10 should be n the introduction.
Response 3: Figure 10 was introduced in the mentioned section as Figure 1.
Point 4: Authors are claiming this in the conclusion. "These very promising results suggests that BrCl-flav is in fact a compound 745 with a potent antibacterial activity against ESKAPE pathogens, which may be used to develop new effective antimicrobial agents able to bypass bacterial multidrug resistance."
I don't see antimicrobial data against all six bacteria of ESKAPE pathogens. So, the above statement is not true.
Response 4: It is true that not all the ESKAPE strains were extensively tested in our work. However, we provide data concerning the MIC and MBC for all ESKAPE pathogens used in our study, therefore we may suggest that BrCl-flav is a promising agent against the above-mentioned pathogens. To avoid confusion, next sentence was modified:
These very promising results suggests that BrCl-flav is in fact a compound with a potent antibacterial activity against representative ESKAPE pathogens, which may be used to develop new effective antimicrobial agents able to bypass bacterial multidrug resistance.
Round 2
Reviewer 3 Report
I recommend the manuscript for publication in Antibiotics.